# BRCA1/2 Variants and Metabolic Factors: Results From a Cohort of Italian Female Carriers

**DOI:** 10.3390/cancers12123584

**Published:** 2020-11-30

**Authors:** Andreina Oliverio, Eleonora Bruno, Mara Colombo, Angelo Paradiso, Stefania Tommasi, Antonella Daniele, Daniela Andreina Terribile, Stefano Magno, Donatella Guarino, Siranoush Manoukian, Bernard Peissel, Paolo Radice, Patrizia Pasanisi

**Affiliations:** 1Epidemiology and Prevention Unit, Fondazione IRCCS Istituto Nazionale dei Tumori di Milano, 20133 Milan, Italy; andreina.oliverio@istitutotumori.mi.it (A.O.); eleonora.bruno@istitutotumori.mi.it (E.B.); patrizia.pasanisi@istitutotumori.mi.it (P.P.); 2Unit of Molecular Bases of Genetic Risk and Genetic Testing, Fondazione IRCCS Istituto Nazionale dei Tumori di Milano, 20133 Milan, Italy; mara.colombo@istiutotumori.mi.it; 3Experimental Oncology, Center for Study of Heredo-Familial Tumors, IRCCS Istituto Tumori “Giovanni Paolo II” Bari, 70124 Bari, Italy; a.paradiso@oncologico.bari.it (A.P.); antonella.daniele@oncologico.bari.it (A.D.); 4Molecular Diagnostics and Pharmacogenetics Unit, IRCCS Istituto Tumori “Giovanni Paolo II” Bari, 70124 Bari, Italy; s.tommasi@oncologico.bari.it; 5Università Cattolica S. Cuore, 00168 Rome, Italy; danielaandreina.terribile@policlinicogemelli.it (D.A.T.); terapie.integrate@policlinicogemelli.it (D.G.); 6Fondazione Policlinico Universitario A. Gemelli IRCCS, 00168 Rome, Italy; stefano.magno@policlinicogemelli.it; 7Unit of Medical Genetics, Fondazione IRCCS Istituto Nazionale dei Tumori di Milano, 20133 Milan, Italy; siranoush.manoukian@istitutotumori.mi.it (S.M.); bernard.peissel@istitutotumori.mi.it (B.P.)

**Keywords:** BRCA genes, pathogenic variants, BRCA-related cancer, metabolic factors

## Abstract

**Simple Summary:**

Women who inherit a BRCAmutation face a high lifetime risk of developing cancer. However, several factors, genetic and/or “environmental”, may influence BRCA penetrance. We studied in 438 women carriers of BRCA1/2 the association of metabolic factors with BRCA1/2 variants and the risk effect of metabolic exposures in relation to the position of the mutations within the BRCA1/2. The pathogenic variants were divided into loss of function (LOF) and nonsynonymous variants. Findings from this study suggest that higher insulin levels are significantly associated with BRCA LOF variants compared to nonsynonymous variant carriers. Therefore, our results support the hypothesis that the impairment of BRCA protein functions could result in a different association with “metabolic” factors, possibly due to a genetic effect on the etiology of altered response to metabolism.

**Abstract:**

Women carriers of pathogenic variants (mutations) in the *BRCA1/2* genes face a high lifetime risk of developing breast cancer (BC) and/or ovarian cancer (OC). However, metabolic factors may influence BRCA penetrance. We studied the association of metabolic factors with *BRCA1/2* variants and the risk effect of metabolic exposures in relation to the position of the mutations within the *BRCA1/2*. Overall, 438 women carriers of *BRCA1/2* mutations, aged 18–70, with or without a previous diagnosis of BC/OC and without metastases, who joined our randomized dietary trial, were included in the study. The pathogenic variants were divided, according to their predicted effect, into loss of function (LOF) and nonsynonymous variants. The association between metabolic exposures and variants were analyzed by a logistic regression model. LOF variant carriers showed higher levels of metabolic parameters compared to carriers of nonsynonymous variants. LOF variant carriers had significantly higher levels of plasma glucose and serum insulin than nonsynonymous variant carriers (*p* = 0.03 and *p* < 0.001, respectively). This study suggests that higher insulin levels are significantly associated with LOF variants. Further investigations are required to explore the association of metabolic factors with LOF variants and the mechanisms by which these factors may affect BRCA-related cancer risk.

## 1. Introduction

Women who have inherited pathogenic variants (mutations) in the *BRCA1* and *BRCA2* genes (*BRCA1/2*) face a very high lifetime risk of developing breast cancer (BC) and/or ovarian cancer (OC). A recent prospective study, including approximately 10.000 *BRCA1/2* pathogenic variant carriers, observed a chance of developing BC by age 80 of 72% (95% CI, 65–79%) for *BRCA1* and of 69% (95% CI, 61–77%) for *BRCA2* mutation carriers, respectively. For OC, the lifetime risk at age 80 years was 44% (95% CI, 36–55%) for *BRCA1* and 17% (95% CI, 11–25%) for *BRCA2* mutation carriers [1].

BRCA1/2 proteins have several functional domains that bind to specific partners [2]. BRCA1 contains an N-terminal RING domain that has E3 ubiquitin ligase activity (which catalyzes protein ubiquitylation), a nuclear export sequence (NES), three nuclear localization signals (NLSs) in the central portion (which facilitates nuclear import of BRCA1), a coiled-coil domain that binds the N-terminal of the partner and localizers of BRCA2 (PALB2) protein, and a C-terminal BRCT domain that binds proteins phosphorylated by the Ataxia telangiectasia mutated (ATM) and Ataxia telangiectasia and Rad3-related (ATR) kinases [3]. BRCA2 contains an N-terminal domain that interacts with the C-terminus of PALB2; eight BRC repeats that bind the RAD51 protein, and a C-terminal DNA-binding domain (DBD) that binds single-strand DNA (ssDNA) and double-strand DNA (dsDNA) [4]. The DBD contains five components: a 190-amino-acid α-helical domain, three oligonucleotide-binding (OB1, OB2, OB3) folds that are ssDNA-binding modules, and a tower domain (TD) that protrudes from OB2 and binds dsDNA. The α-helical domain, OB1 and OB2 also associated with the deleted in split-hand/split-foot syndrome (DSS1) protein, which has been linked with BRCA2 protein stabilization [5].

The Consortium of Investigators of Modifiers of *BRCA 1/2* (CIMBA) reported that the risk of BC/OC significantly differs on the basis of the location of the specific mutations [6]. In female carriers, mutations in the RING and BRCT domains of *BRCA1* were associated with higher BC risk than mutations located in other gene regions. As for *BRCA2*, mutations mapped to the BRC domains showed increased OC risk while mutations occurring at the 5′ and 3′ regions, corresponding approximately to the PALB2-binding and DBD domains, respectively, were associated with increased BC risk.

BRCA gene testing is provided not only for disease diagnosis but also for cancer treatment and follow-up and for disease prevention. In this context, it has been shown that the penetrance of BRCA mutations may be modulated through other genetic as well as “environmental” factors [7]. Previous studies reported an increase in BRCA mutation penetrance with greater body weight, especially in *BRCA1* mutation carriers. Obesity may increase BRCA mutation penetrance through a number of mechanisms, including insulin and insulin-like growth factor I (IGF-I) regulation [8,9]. Consistently, retrospective case–control analyses suggested that IGF-I and metabolic syndrome (MS) are associated with an increased BRCA mutation penetrance [8,10].

We have conducted a randomized controlled trial [11,12,13,14,15] in women with BRCA mutations to test whether a Mediterranean dietary intervention with moderate protein restriction significantly reduces IGF-I and other potential modulators of penetrance. A cross-sectional analysis of the baseline participants’ metabolic parameters suggested that higher fat mass and the presence of four or five factors of the metabolic syndrome were significantly associated with the risk of BRCA-related cancer [15].

In the present study, we analyzed the association of metabolic factors with *BRCA1/2* pathogenic variants at baseline. In particular, we focused on the risk effect of metabolic exposures in relation to the position of the mutations within the *BRCA1/2* genes. The study included 483 female carriers of *BRCA1/2* pathogenic variants with complete genetic testing who joined our dietary randomized controlled trial.

## 2. Results

A complete genetic test result was available for 438 out of the 502 women who joined the dietary intervention trial. Among these carriers, 269 (61.4%) had mutations in *BRCA1* and 169 (38.6%) in *BRCA2*; 163 women (37.2%) had frameshift deletions, 99 (22.6%) had nonsense variants, 89 (20.3%) had frameshift insertions, 49 (11.3%) had missense variants, 14 (3.2%) had spliceogenic variants, 14 (3.2%) had small in-frame deletions, 5 (1.1%) had frameshift substitution (i.e., variants introducing a frameshift in the gene open reading frame as the result of the deletion of one or more nucleotides replaced by a variable number of newly inserted nucleotides) and 5 (1.1%) had large deletion. Based on these observations, 375 (85.6%) women were classified as carriers of LOF variants and 63 (14.4%) women as carriers of nonsynonymous variants (Table 1).

The general characteristics of the study population by *BRCA1/2* are reported in Table 1. Among *BRCA1* women, those with nonsynonymous variants (17.1%) were significantly more represented than in *BRCA2* women (10.1%) (*p* = 0.004). As for mutation position, carriers with *BRCA1* LOF variants (82.9%) were distributed as follows: 2.2% had mutations in the 5′-end group, 46.1% in the inner group and 34.6% in the 3′-end group. Among the carriers of *BRCA2* LOF variants (89.9%), the distribution was as follows: 1.8% had mutations in the 5′-end group, 46.7% in the inner group and 41.4% in the 3′-end-group.

The distribution of reproductive factors was fairly similar in *BRCA1* and *BRCA2* mutation carriers.

As for disease status, 215 (82.1%) women had a previous diagnosis of BC, 36 (13.7%) of OC, 11 (4.2%) had both BC and OC, and 176 were unaffected. Among the affected women, 163 had *BRCA1* mutations, and 99 had *BRCA2* mutations. Among BC cases, women with *BRCA1* mutations showed a significantly higher frequency of infiltrating ductal carcinomas (*p* = 0.01), ER-negative tumors (*p* < 0.001) and tumors with Ki-67 > 14 (*p* = 0.005). Women with *BRCA2* mutations had a significantly higher frequency of axillary node metastases (*p* = 0.02).

The distribution of metabolic factors was comparable in *BRCA1* and *BRCA2* mutation carriers (Appendix A).

As regards the association with overall cancer risk (Table 2), the OR of LOF variant carriers versus nonsynonymous variant carriers was 1.53 (95% CI: 0.80–2.92) and 1.14 (95% CI: 0.40–3.22) for *BRCA1* and *BRCA2*, respectively. The OR of BC in LOF variant carriers versus nonsynonymous variant carriers was 1.42 (95% CI: 0.71–2.82) and 0.90 (95% CI: 0.31–2.66) for *BRCA1* and *BRCA2*, respectively.

The metabolic characteristics by variant type are set out in Table 3. Plasma glucose and insulin were significantly higher in carriers of LOF variants than in carriers of nonsynonymous variants (*p* = 0.03 and *p* < 0.001, respectively). The latter showed a slightly worse body composition and higher levels of triglycerides and IGF-I compared to carriers of LOF variants.

Evaluating the association between metabolic characteristics and variant type by a logistic regression model (Table 4), LOF variant carriers showed higher levels of metabolic parameters compared to carriers of nonsynonymous variants, yet without any significant result. In the restricted analysis including only BC affected and unaffected women, LOF variant carriers had a significant OR of 2.42 (95% CI: 1.12–5.23) in the upper tertile of insulin concentration compared to carriers of nonsynonymous variants. In the stratified analysis by disease status, higher insulin levels were associated with LOF variants both in unaffected and BC-affected women.

Table 5 reports the associations of BRCA-related cancer risk with variant type and position in *BRCA1* and *BRCA2* women. No significant associations were detected, although a general trend to higher risk, both overall and BC specific, was observed in carriers of *BRCA1* LOF variants.

Table 6 reported the distribution of metabolic characteristics by variant type and position. In *BRCA1*, women with mutations located at the 3′-end showed significantly higher levels of insulin compared to carriers of nonsynonymous variants (*p* = 0.02). In *BRCA2*, women with mutations in the inner region and at the 3′-end showed, compared to carriers of nonsynonymous variants, significantly higher levels of plasma glucose (*p* = 0.01 for both groups), insulin (*p* = 0.01 for both groups) and diastolic pressure (*p* = 0.02 and *p* = 0.01, respectively)

## 3. Discussion

In the present study, we analyzed the association of BRCA-related cancer risk and metabolic factors with *BRCA1/2* variant type and position in a cohort of 438 female carriers who joined our dietary randomized controlled trial.

Previous studies suggested that nonsynonymous variants of BRCA genes may be associated with a reduced risk of BC in comparison to LOF variants [16,17,18]. In addition, BC risk was reported to be significantly higher in women with mutations in the RING and BRCT domains of *BRCA1* and lower in women with mutations in the BRC repeats of *BRCA2* [6]. However, the mechanisms responsible for the observed differences remain at present undetermined.

Our results are partially consistent with the above observations, although at a not significant level, most likely due to the reduced size of the study population. In particular, as concerned BC risk, we observed an associated OR of 1.42 for *BRCA1* LOF variants compared to nonsynonymous variants. Moreover, when considering the position of LOF variants within the gene, those mapped at the 5′-end of BRCA1, which lead to loss of all functional protein domains, showed a higher BC risk compared to LOF variants located downstream.

Interestingly, findings from this study suggest that higher insulin levels are significantly associated with BRCA LOF variants. Moreover, this association seemed stronger in *BRCA2* women who, in addition, showed a worse metabolic condition compared to nonsynonymous variant carriers. Therefore, our results support the hypothesis that the impairment of BRCA protein functions could result in a different association with “metabolic” factors, possibly due to a genetic effect on the etiology of altered response to metabolism.

The association of metabolic factors with BC risk in both the general population and BRCA mutation carriers has been widely documented. Insulin levels and plasma glucose have been associated with a higher risk of sporadic BC [19,20]. More recently, metabolic dysfunction and obesity have been shown to be a stronger predictor of BC risk than obesity alone [21,22]. Metabolic dysfunction is also associated with hereditary BC [8], and female carriers of BRCA mutations more frequently develop type-2 diabetes after a BC diagnosis [9]. In addition, the pre-diabetic condition, when the levels of insulin are typically very high, increased the risk of BC in BRCA-mutant women [9].

Insulin may be considered as a growth factor that stimulates cell mitosis, cell migration and inhibits apoptosis, thus promoting cancer growth [23]. These proliferative effects involve the activation of Ras and the mitogen-activated protein kinase pathway [24]. Furthermore, insulin inhibits the liver production of sex hormone-binding globulin (SHBG), thus increasing the bioavailability of estrogen [25]. Since BRCA mutations confer a lower ability to repair DNA damage, mutation carriers may be more sensitive to the insulin mitogenic effect. Further larger prospective studies are needed to explore the association between insulin and altered metabolism with BRCA gene LOF variants and mechanisms by which the effects of these factors may affect BRCA-related cancer risk.

There are several limitations to this study. In addition to the relatively small number of recruited cases, it must be acknowledged that data and blood samples were collected when the women with *BRCA* mutations were enrolled into the dietary trial and that both affected and unaffected volunteers were recruited. Compared to healthy women, the metabolic parameters of the affected women may be influenced by cancer treatments in the years before entering the trial. Hormonal treatments after cancer, for example, may worsen a woman’s body composition and metabolism. However, the stratified analysis by disease status showed that higher insulin levels were associated with LOF variants both in affected and unaffected women.

We expect that the prospective follow-up of the trial cohort will provide further insights into the interaction between BRCA variants and metabolic factors. This would provide a rationale in support of dietary interventions as personalized risk reduction measures for mutation carriers.

## 4. Materials and Methods

### 4.1. Study Subjects

The Italian multicenter, prospective, randomized controlled trial (NCT03066856) aimed at investigating the effect of the adoption of the Mediterranean diet with moderate protein restriction in reducing IGF-I levels, body weight and other metabolic modulators of BRCA mutation penetrance. Detailed information regarding trial design and main results have already been reported [11,12,13,14,15].

Following the intake criteria of the above trial, study subjects were recruited through family clinics and patient associations among women (18–70 years) carriers of *BRCA1/2* mutations, following complete gene testing (sequencing of all coding regions and intron–exon boundaries, followed by Multiplex Ligation-dependent Probe Amplification assay for detection of large insertions/deletions), with or without a previous diagnosis of BC/OC and without clinical evidence of metastases. Unaffected women who underwent bilateral prophylactic mastectomy were excluded from the cohort. All participants signed informed consent for study participation. The study was approved by the Ethics Committee of the coordinating center, the Fondazione IRCCS Istituto Nazionale dei Tumori (INT) di Milano, Italy (approval number: INT106/13).

### 4.2. Data Collection

Participants Were Requested:to provide information on their *BRCA1/2* mutational status and, whenever possible, provide the complete report of the genetic test;to complete a questionnaire on their medical history and major cancer risk factors, including menstrual history, reproductive and behavioral factors;to attend a clinic examination for anthropometric and body composition measurements;to provide a 20 mL blood sample for metabolic and hormonal assays;to provide information on their health status, and to allow the study officials to contact their usual physicians, to consult clinical notes and to examine biopsy material, as necessary;to complete a 24-h food frequency diary of the previous day and the validated 14-point MEDAS questionnaire [26].

Anthropometric measurements, blood samples and dietary data were provided at baseline and at the end of a six-month dietary intervention.

### 4.3. Laboratory Methods

The collected blood samples were stored at −80 °C. Plasma glucose, triglycerides, total LDL and HDL cholesterol, were measured using routine clinical laboratory techniques. Serum IGF-I (Biosource-Nivelles, Belgium) and insulin (Immunotech-Prague, Czech Republic) were measured using commercial kits as described [15].

All tests were performed blinded to the participant’s disease status.

### 4.4. Nomenclature and Classification of Variants

The description of the genetic variants followed the Human Genome Organization-approved nomenclature system of the Human Genome Variation Society (hg18) (http://varnomen.hgvs.org/). Genetic variants were analyzed by comparison with a consensus reference sequence of each gene (NM_007294.3 for *BRCA1* and NM_000059.3 for *BRCA2*).

The ClinVar (https://www.ncbi.nlm.nih.gov/clinvar/) and BRCA exchange (https://brcaexchange.org/) databases and recent scientific literature were used to inform clinical classification of variants. According to the IARC guidelines [27], we defined five classes of variants, i.e., pathogenic [C5], likely pathogenic [C4], benign [C1], likely benign [C2] and variants of uncertain significance [VUS; C3]. Variant classification was performed following the criteria of the Evidence-based Network for the Interpretation of Germline Mutant Alleles (ENIGMA), an international consortium of investigators aimed at determining the clinical significance of sequence variants in BC/OC genes (http://www.enigmaconsortium.org/; Version 2.5.1 29 June 2017). In the study, only C4 and C5 variants, cumulatively defined as “pathogenic”, were considered.

The pathogenic variants were divided into two groups, according to their predicted effect:loss of function (LOF) variants, including frameshift, nonsense, large deletion and spliceogenic variants, introducing a premature termination codon (PTC) or leading to the in-frame loss of gene regions coding for functional domains;nonsynonymous variants, including missense and small in -rame deletion.

The above variants were further subdivided based on their location within the genes and taking into account the addressed functional domains. To this aim, we followed the catalog of BRCA1 and BRCA2 conserved domains/motifs as reported in the ENIGMA *BRCA1/2* gene variant classification criteria.

In the present analysis, the considered functional domains/motifs were, from N-terminus to C-terminus: (1) RING, (2) NES, NSL and coiled-coil, and (3) BRCT domains in BRCA1; (1) PALB2-binding, (2) BRC repeats, and (3) DNA-binding domains in BRCA2. As regards LOF variants, mutation carriers were subdivided into three groups: (a) carriers of 5′-end mutations, i.e., located upstream the distal boundary of the first (N-terminal) domain and, therefore, putatively affecting all functional domains; (b) carriers of “inner” mutations located downstream the N-terminal domain and upstream the distal boundary of the second (inner) domain, putatively affecting this and the C-terminal domain(s); (c) carriers of 3′-end mutations located downstream the distal boundary of the inner domain, thus putatively affecting only the C-terminal domain(s). Carriers of nonsynonymous variants served as a reference group for the risk analyses.

### 4.5. Statistical Analysis

General characteristics of the study population were summarized by *BRCA1/2* using frequencies or means and standard deviation (SD) and compared using χ2 or *t*-tests, as appropriate.

The odds ratios (ORs) and 95% confidence interval (CI) of BRCA-related cancer risk by variant type and position were evaluated by a logistic regression model after controlling for the study center.

Women’s metabolic characteristics were summarized by variant types using means and standard error of the mean (SEM) and compared with ANOVA controlling for age (tertiles), study center, menstrual status (still menstruated, natural menopause, induced menopause), BMI (tertiles), and mutation (*BRCA1/2*). The association between metabolic exposures and variant type was analyzed by a logistic regression model after controlling for the same adjustments. The cutoff for the metabolic variables was chosen based on MS [28]. The associations between metabolic exposures and variant positions were evaluated by ANOVA. A *p*-value of < 0.05 was taken as significant. All statistical tests were two-sided. Analyses were done using STATA 14 (StataCorp–College Station, TX, USA) statistical package.

## 5. Conclusions

This study suggests that higher insulin levels are significantly associated with LOF variants. Further investigations are required, and we expect that the prospective follow-up of the trial cohort will provide further insights into the interaction between BRCA variants and metabolic factors.

## Figures and Tables

**Table 1 cancers-12-03584-t001:** Baseline characteristics of the study population by *BRCA1* and *BRCA2* genes (*BRCA1/2)* in 438 female carriers.

Characteristics		Total Population (438)	BRCA1 (269)	BRCA2 (169)	*p* Value *
Age (years)		46.9 ± 11.1	47.2 ± 10.8	46.6 ± 11.5	0.53
Pathogenic variants (%) LOF Nonsynonymous		85.6 14.4	82.9 17.1	89.9 10.1	0.004
BRCA1 variant group (%)		N/A	17.1 2.2 46.1 34.6	N/A	
NONSYNONYMOUS					
LOF	5′-end				
	Inner				
	3′-end				
BRCA2 variant group (%) NONSYNONYMOUS		N/A	N/A	10.1 1.8 46.7 41.4	
LOF	5′-end				
	Inner				
	3′-end				
Education (%)					
First level Second level Third level		16.7 44.5 38.8	17.1 44.6 38.3	16.0 44.4 39.6	0.94
Menarche (yrs)		12.4 ± 1.4	12.3 ± 1.5	12.4 ± 1.4	0.43
Age at first live birth (yrs)		29.0 ± 4.9	28.7 ± 4.9	29.4 ± 4.8	0.21
Pregnancy (yes) (%)		71.0	73.2	66.5	0.20
Number of children (%) 1 ≥2		30.7 69.3	28.6 71.4	34.3 65.7	0.31
Menopause (%)		72.9	72.5	73.4	0.73
Natural menopause (%)		25.1	23.6	27.4	0.44
Oral contraceptive use in the past (%)		67.8	66.5	70.0	0.47
Smoke in the past (%)		27.3	27.5	27.0	0.79
		Total affected (262)	BRCA 1 (163)	BRCA 2 (99)	
Cancer type (% if affected) Breast Breast and ovarian Ovarian		82.1 4.2 13.7	79.1 4.9 16.0	86.9 3.0 10.1	0.29
Age at diagnosis (years)		44.0 ± 8.7	43.5 ± 9.1	44.7 ± 8.1	0.31
		Breast cancer (215)	BRCA1 (129)	BRCA2 (86)	
Infiltrating duct (%) ER-negative (%) Axillary node metastasis (%) Ki-67 > 14 (%)		86.0 44.6 31.6 87.2	88.1 58.8 25.6 94.0	82.9 24.1 40.7 78.5	0.01 <0.001 0.02 0.005

* *p* values were established using the chi-squared test for categorical variables and Student’s t-test for continuous variables.

**Table 2 cancers-12-03584-t002:** Association between BRCA-related cancer risk and variant type.

All Cancers	BRCA1 163 Affected and 106 Unaffected	BRCA2 99 Affected and 70 Unaffected
	OR (95% CI) *	OR (95% CI) *
Nonsynonymous LOF	1 1.53 (0.80–2.92)	1 1.14 (0.40–3.22)
Breast Cancer	**BRCA1** **129 affected and 106 unaffected**	**BRCA2** **86 affected and 70 unaffected**
	**OR (95% CI) ***	**OR (95% CI) ***
Nonsynonymous LOF	1 1.42 (0.71–2.82)	1 0.90 (0.31–2.66)

* adjusted for study center.

**Table 3 cancers-12-03584-t003:** Metabolic characteristics by BRCA variant types in 438 female carriers.

Metabolic Characteristics	Nonsynonymous (63)	LOF (375)	*P* *
Weight (kg)	67.0 (2.2)	64.2 (0.7)	0.18
BMI (kg/m^2^)	25.7 (0.8)	24.5 (0.3)	0.13
Waist circumference (cm)	78.7 (1.4)	78.6 (0.7)	0.93
Hip circumference (cm)	101.3 (1.1)	99.9 (0.5)	0.39
Waist to height ratio (cm/cm)	0.49 (0.01)	0.49 (0.01)	0.79
Fat mass (%)	30.6 (1.0)	31.5 (0.5)	0.65
Systolic pressure (mmHg)	122.0 (1.9)	125.5 (0.9)	0.06
Diastolic pressure (mmHg)	81.0 (1.4)	81.6 (0.6)	0.56
Glycemia (mg/dL)	97.2 (2.7)	101.5 (1.2)	0.03
Total cholesterol (mg/dL)	197.5 (4.3)	200.1 (2.1)	0.81
HDL cholesterol (mg/dL)	67.8 (2.2)	67.9 (0.9)	0.90
LDL cholesterol (mg/dL)	116.0 (4.0)	119.0 (1.9)	0.96
Triglycerides (mg/dL)	107.8 (13.4)	103.2 (2.8)	0.46
IGF-I (ng/mL)	182.6 (7.7)	177.9(3.6)	0.66
Insulin (µIU/mL)	16.5 (1.3)	21.2 (1.0)	<0.001

* *p*: of ANOVA controlling for age (tertiles), study center, menstrual status (regular menstrual cycle, natural menopause, iatrogenic menopause), BMI (tertiles) and mutation (BRCA1/2).

**Table 4 cancers-12-03584-t004:** Association between metabolic characteristics and BRCA variant type.

Metabolic Characteristics	Total Study Population 375 LOF vs. 63 Nonsynonymous	BC Affected and Unaffected 332 LOF vs. 59 Nonsynonymous
BMI (kg/m^2^)	**OR (95% CI) ***	**OR (95% CI) ***
≤23.4 >23.4	1 0.66 (0.37–1.15)	1 0.62 (0.35–1.11)
Fat mass (%)		
<30 ≥30	1 1.01 (0.55–1.83)	1 1.05 (0.57–1.93)
Waist to height ratio (cm/cm)		
≤0.50 >0.50	1 1.34 (0.70–2.55)	1 1.41 (0.74–2.69)
Waist circumference (cm)		
<85 ≥85	1 1.10 (0.55–2.15)	1 1.20 (0.60–2.39)
Glycemia (mg/dL)		
<110 ≥110	1 1.54 (0.80–2.97)	1 1.75 (0.90–3.42)
HDL cholesterol (mg/dL)		
<50 ≥50	1 1.45 (0.69–3.60)	1 1.58 (0.59–4.23)
Triglycerides (mg/dL)		
<150 ≥150	1 1.07 (0.47–2.46)	1 1.21 (0.51–2.89)
Blood pressure (mmHg)		
<130/85 ≥130/85	1 1.43 (0.80–2.49)	1 1.57 (0.87–2.83)
Insulin (µIU/mL)		
1 (1.0–10.5) 2 (10.6–23.3) 3 (23.4–101.2)	1 1.26 (0.64–1.54) 1.91 (0.90–4.03)	1 1.41 (0.70–2.86) 2.42 (1.12–5.23)
IGF-I (ng/mL)		
1 (60.3–140.7) 2 (140.8–199.5) 3 (199.6–508.1)	1 0.63 (0.31–1.29) 1.07 (0.47–2.43)	1 0.48 (0.23–1.02) 0.72 (0.33–1.57)

* odds ratios (ORs) adjusted for age (tertiles), center, and menstrual status (regular menstrual cycle, natural menopause, iatrogenic menopause), BMI (tertiles), study center and mutation (BRCA1/2).

**Table 5 cancers-12-03584-t005:** Association between BRCA-related cancer risk and the variant type and position.

Variant Position		Number (*N*)	BRCA-Related Cancer OR * (95% CI)			BRCA–Related BC OR * (95% CI)		
BRCA1		N Total 269	N Affected 163		*p*	N Affected 129		*p*
Nonsynonymous		46 6 124 93	24 5 76 58	1 4.49 (0.46–44.1) 1.32 (0.65–2.70) 1.44 (0.65–3.16)	0.20 0.44 0.37	20 5 58 46	1 4.97 (0.51–48.4) 1.25 (0.59–2.60) 1.30 (0.56–2.98)	0.19 0.56 0.54
LOF	5′-end Inner 3′-end							
BRCA2		169	99	OR (95% CI)	*p*	86	OR (95% CI)	*p*
Nonsynonymous		17 3 79 70	10 3 44 42	1 NA 1.09 (0.33–3.60) 1.27(0.39–4.16)	0.88 0.69	10 3 34 39	1 NA 0.86 (0.26–2.82) 1.07 (0.33–3.55)	0.78 0.91
LOF	5′-end Inner 3′-end							

* ORs adjusted for study center.

**Table 6 cancers-12-03584-t006:** Distribution of metabolic characteristics by BRCA variant type and position.

Metabolic Characteristics	BRCA1 (269)				BRCA2 (169)			
	Nonsynonymous (46)	LOF-5′ End (6)	LOF-Inner (124)	LOF-3′ End (93)	Nonsynonymous (17)	LOF-5′ End (3)	LOF-Inner (79)	LOF-3′ End (70)
Weight (kg)	65.8 (2.0)	66.0 (5.7)	63.9 (1.2)	65.8 (1.4)	70.5 (3.3)	61.1 (7.9)	64.8 (1.5)	61.9 (1.6)
BMI (kg/m^2^)	25.4 (0.8)	25.1 (2.2)	24.3 (0.5)	25.2 (0.6)	26.4 (1.2)	23.1 (2.9)	24.9 (0.6)	23.5 (0.6)
Waist circumference (cm)	76.8 (1.8)	79.9 (4.9)	78.3 (1.1)	80.1 (1.2)	83.5 (3.5)	74.5 (8.3)	80.1 (1.6)	75.6 (1.7)
Hip circumference (cm)	100.1 (1.5)	100.8 (4.2)	100.1 (0.9)	100.9 (1.1)	104.3 (2.4)	101.2 (5.7)	99.8 (1.1)	98.2 (1.2)
Waist to height ratio (cm/cm)	0.48 (0.01)	0.49 (0.03)	0.48 (0.01)	0.50 (0.01)	0.51 (0.02)	0.46 (0.05)	0.50 (0.01)	0.47 (0.01)
Fat mass (%)	29.7 (1.2)	33.4 (3.4)	30.5 (0.8)	33.5 (0.8)	33.4 (2.3)	38.0 (4.9)	31.5 (1.0)	29.8 (1.1)
Systolic pressure (mmHg)	122.0 (2.4)	121.8 (6.6)	127.8 (1.5)	123.7 (1.8)	122.1 (3.8)	110.0 (15.6)	124.8 (1.8)	124.9 (1.9)
Diastolic pressure (mmHg)	82.2 (1.5)	83.2 (4.2)	82.9 (0.9)	80.3 (1.1)	78.1 (2.6)	90.0 (10.7)	81.4 (1.2) *	80.9 (1.3) **
Glycemia (mg/dL)	101.1 (3.2)	96.5 (8.8)	104.2 (1.9)	98.9 (2.2)	86.6 (5.7)	94.0 (13.6)	99.5 (2.6) **	103.3 (2.8) **
Total cholesterol (mg/dL)	198.6 (5.7)	196.7 (15.8)	203.2 (3.5)	201.0 (4.0)	194.5 (9.6)	194.3 (22.8)	198.2 (4.4)	195.9 (4.7)
HDL cholesterol (mg/dL)	68.4 (2.4)	60.2 (6.6)	69.9.3 (1.5)	66.3 (1.7)	66.2 (4.2)	83.7 (10.1)	62.1 (1.9)	72.4 (2.1)
LDL cholesterol (mg/dL)	115.9 (5.2)	124.3 (14.3)	119.0 (3.1)	123.3 (3.6)	116.2 (9.1)	105.3 (21.6)	122.3 (4.2)	109.8 (4.5)
Triglycerides (mg/dL)	109.4 (10.6)	98.5 (29.5)	105.6 (6.5)	105.5 (7.5)	102.1 (12.0)	61.3 (28.6)	108.3 (5.6)	92.3 (5.9)
IGF-I (ng/mL)	186.8 (8.4)	136.5 (25.4)	182.5 (5.2)	175.6 (6.0)	170.6 (17.1)	139.9 (39.5)	178.7 (7.8)	176.6 (8.3)
Insulin (µIU/mL)	17.9 (2.7)	24.5 (8.3)	22.5 (1.7)	19.6 (1.9)*	12.6 (4.6)	21.3 (10.7)	19.7 (2.1) **	22.3 (2.3) **

* *p* = 0.02; ** *p* = 0.01 (ANOVA, significant results of comparison).

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
