# Peer review of "BRCA1/2 Variants and Metabolic Factors: Results From a Cohort of Italian Female Carriers"

_cancers, 2020, doi:10.3390/cancers12123584_

Round 1

Reviewer 1 Report

Comments for the authors

Oliverio et al investigate the effects of metabolic factors on breast cancer risk in 438 BRCA1/2 pathogenic variant carriers. They found that women with loss-of-function BRCA1/2 variants had significantly higher levels of insulin in their blood than women with non-truncating (missense or in frame deletions) and that loss-of-function variant carriers with high insulin levels had a higher risk of developing breast cancer than women with a non-truncating variant and high insulin levels. Other measures of metabolic function were not significantly different.

I have only minor comments

It should be explained why women with metastases were excluded from the study.

Some terminology should be altered e.g. line 50 "mutated female individuals" should be changed to "BRCA1/2 pathogenic variant carriers".

Line 80 "four or five metabolic risk factors". Can you be more specific here?

It is not clear what is meant by "frameshift substitution".

The manuscript could use some further editing for use of English.

Author Response

We think that our manuscript has been improved thanks to the comments and suggestions made by the reviewers, and we trust our work is now acceptable for publication.

We changed the manuscript according to the minor revisions requested.

All the manuscript changes are reported in red.

Comments for the authors

Oliverio et al investigate the effects of metabolic factors on breast cancer risk in 438 BRCA1/2 pathogenic variant carriers. They found that women with loss-of-function BRCA1/2 variants had significantly higher levels of insulin in their blood than women with non-truncating (missense or in frame deletions) and that loss-of-function variant carriers with high insulin levels had a higher risk of developing breast cancer than women with a non-truncating variant and high insulin levels. Other measures of metabolic function were not significantly different.

I have only minor comments

It should be explained why women with metastases were excluded from the study.

-We thank the reviewer for the time he/she dedicated to our paper. The women included into the present study joined our dietary intervention trial aimed at reducing IGF-I and other modulators of BRCA mutation penetrance,and were recruited following the trial intake criteria (see line 216). According to the trial protocol, the cohort will be prospectively followed-up to investigate the association between the metabolic factors under study (and modified by intervention) and the occurrence of new primary cancers or cancer recurrences. Therefore, we excluded “a priori” from the trial cohort the women who have already experienced an important progression of the disease.

Some terminology should be altered e.g. line 50 "mutated female individuals" should be changed to "BRCA1/2 pathogenic variant carriers".

-We agree and we changed the sentences accordingly (see lines 52, 90 and 162).

Line 80 "four or five metabolic risk factors". Can you be more specific here?

-We agree and we changed the sentence to “four of five factors of the metabolic syndrome” (see line 86-87). We also added the reference of our cross-sectional analysis currently in press (Ref. 15).

It is not clear what is meant by "frameshift substitution".

-With the term ‘frameshift substitution” we referred to variants introducing a frameshift in the gene open reading frame as the result of the deletion of one or more nucleotides replaced by a variable number of newly inserted nucleotides. This explanation has been added in the manuscript (see lines 97-99)

The manuscript could use some further editing for use of English.

-We agree and we improved the English editing

Reviewer 2 Report

The authors have constructed a well-written and concise manuscript that I believe should be published, provided certain things are addressed. First, visual inspection of the data is unimpressive with regards to showing differences, which I am guessing relates to utilizing the SD in tables.  This may be correctable by using the SEM.  As it stands the reader has to accept that testing per se detected significance while looking at values that do not appear meaningfully significant. Second, considering only the metabolic parameters (insulin and glycemia), to what extent can the difference have physiologic significance? 

The attention to detail included in the manuscript is exceptional. However, since the details are extensive, the lengthy tables presented mute the reader's opportunity to see the significant differences pointed out through the author's use of bolded font.  This issue may be solvable by using a colored font, especially in Table 6, and Tables 1 & 3 as well. I applaud the authors for highlighting that a larger study group is needed to address their problem.  They should use their current results to perform a power analysis to state how large the study group would need to be.

I feel the recommendations and conditions that my review mandates are reasonable, not difficult, and do not require additional experimental inspection.

Author Response

We changed the manuscript according to the minor revisions requested.

All the manuscript changes are reported in red.

We would like to respond to reviewer’ suggestions and comments as follows:

Comments and Suggestions for Authors

The authors have constructed a well-written and concise manuscript that I believe should be published, provided certain things are addressed. First, visual inspection of the data is unimpressive with regards to showing differences, which I am guessing relates to utilizing the SD in tables.  This may be correctable by using the SEM. As it stands the reader has to accept that testing per se detected significance while looking at values that do not appear meaningfully significant. Second, considering only the metabolic parameters (insulin and glycemia), to what extent can the difference have physiologic significance? 

-We agree and thank the reviewer for this comment. Accordingly, we included in the tables, SEM instead of SD. Regarding the second question, it is difficult to give a definitive answer at this moment. The higher serum levels of insulin and plasma glucose in carriers of BRCA LOF variants, compared to carriers of nonsynonymous variants might indicate a higher propensity to an insulin resistance condition. This is consistent with the observation that, when we analysed other metabolic factors using the cut offs that define the metabolic syndrome (an insulin resistance syndrome), carriers of LOF variants showed overall a worse metabolic pattern respect to carriers of nonsynonymous variants, although at no significant level. In a baseline case-control analysis of our trial population (now published, see ref. 15), the results suggested that higher fat mass and the presence of a severe dysmetabolism are positively associated with BRCA-related cancer, with a stronger association in women with BRCA2 mutations. The present results on pathogenic variants seem to go to the same direction, enlightening a possible stronger effect of LOF variants. Both findings support the hypothesis that the impairment of BRCA protein functions, particularly in BRCA2 LOF mutation carriers, could result in a different association with “metabolic” factors. The prospective follow-up of our cohort will allow to further investigate these associations with respect to the risk of BRCA-related cancers.

The attention to detail included in the manuscript is exceptional. However, since the details are extensive, the lengthy tables presented mute the reader's opportunity to see the significant differences pointed out through the author's use of bolded font.  This issue may be solvable by using a colored font, especially in Table 6, and Tables 1 & 3 as well. I applaud the authors for highlighting that a larger study group is needed to address their problem.  They should use their current results to perform a power analysis to state how large the study group would need to be.

I feel the recommendations and conditions that my review mandates are reasonable, not difficult, and do not require additional experimental inspection.

-We very thank the reviewer for the time he/she dedicated to our paper and for the positive evaluation of our work. We used coloured fonts in the tables as suggested. As regards the power analysis, we estimated that we were able to find a significant difference in insulin levels between carriers of LOF and nonsynonymous variants with a power of approximately 79%. Increasing the cohort of nonsynonymous variant carriers to approximately 100, our statistical power would raise to 90%. With a total population of 150 nonsynonymous and 500 LOF variant carriers we would probably have enough power also for detecting significant differences in the comparisons among mutations mapped to different gene regions

Reviewer 3 Report

The authors studied in 438 women carriers of BRCA1/2 the association of metabolic factors with BRCA1/2 variants and the risk effect of metabolic exposures in relation with the position of the mutations within BRCA1/2. The pathogenic variants were divided in loss of function (LOF) and nonsynonymous variants and the findings suggest, that higher insulin levels are significantly associated with BRCA LOF variants compared to nonsynonymous variants.

The population investigated was collected for a prospective randomised controlled trial to test, whether a Mediterranean dietary intervention with moderate protein restriction significantly reduces IGF-I and other potential modulators of penetrance. The results are obviously in publication at the moment in a different journal.

In an ideal world one would -in the interest of the reader- prefer a more integrated publication of these important data, which in my opinion belong together. Having said this, I agree that this study is well designed, concisely written, has an important finding and I congratulate the authors.

The study definitely deserves publication in the journal.

Minor comments:

I do not see statistician included in the authors list. The authors may comment on this.

citations with a publication date from 2015 should not be called recently

Careful editing is necessary. Only one example: Discussion, paragraph 4, worst worse 

Author Response

We changed the manuscript according to the minor revisions requested.

All the manuscript changes are reported in red.

We would like to respond to reviewer’ suggestions and comments as follows:

Comments and Suggestions for Authors

The authors studied in 438 women carriers of BRCA1/2 the association of metabolic factors with BRCA1/2 variants and the risk effect of metabolic exposures in relation with the position of the mutations within BRCA1/2. The pathogenic variants were divided in loss of function (LOF) and nonsynonymous variants and the findings suggest, that higher insulin levels are significantly associated with BRCA LOF variants compared to nonsynonymous variants.

The population investigated was collected for a prospective randomised controlled trial to test, whether a Mediterranean dietary intervention with moderate protein restriction significantly reduces IGF-I and other potential modulators of penetrance. The results are obviously in publication at the moment in a different journal.

In an ideal world one would -in the interest of the reader- prefer a more integrated publication of these important data, which in my opinion belong together. Having said this, I agree that this study is well designed, concisely written, has an important finding and I congratulate the authors.

The study definitely deserves publication in the journal.

-We very thank the reviewer for the time he/she dedicated to our paper and for the positive evaluation of our work. We added to this work the reference of our baseline cross-sectional analysis currently in press in Clin Breast Cancer (ref. 15).

Minor comments:

I do not see statistician included in the authors list. The authors may comment on this.

-This was substantially an epidemiological analysis on risk associations. The first author of this paper is an epidemiologist with a PhD in “genetic epidemiology” and we decided to entrust the data analysis to her.

citations with a publication date from 2015 should not be called recently

-We agree and changed the text accordingly (see line 69).

Careful editing is necessary. Only one example: Discussion, paragraph 4, worst worse 

-We agree and performed the English editing.